# Classification of MLH1 Missense VUS Using Protein Structure-Based Deep Learning-Ramachandran Plot-Molecular Dynamics Simulations Method

**DOI:** 10.3390/ijms25020850

**Published:** 2024-01-10

**Authors:** Benjamin Tam, Zixin Qin, Bojin Zhao, Siddharth Sinha, Chon Lok Lei, San Ming Wang

**Affiliations:** 1Ministry of Education Frontiers Science Center for Precision Oncology, Faculty of Health Sciences, University of Macau, Macau SAR, China; 2Cancer Centre, Faculty of Health Sciences, University of Macau, Macau SAR, China; 3Institute of Translational Medicine, Faculty of Health Sciences, University of Macau, Macau SAR, China

**Keywords:** *MLH1*, VUS, deep learning, Ramachandran plot, molecular dynamics simulation, autoencoder, neural network

## Abstract

Pathogenic variation in DNA mismatch repair (MMR) gene *MLH1* is associated with Lynch syndrome (LS), an autosomal dominant hereditary cancer. Of the 3798 *MLH1* germline variants collected in the ClinVar database, 38.7% (1469) were missense variants, of which 81.6% (1199) were classified as Variants of Uncertain Significance (VUS) due to the lack of functional evidence. Further determination of the impact of VUS on *MLH1* function is important for the VUS carriers to take preventive action. We recently developed a protein structure-based method named “Deep Learning-Ramachandran Plot-Molecular Dynamics Simulation (DL-RP-MDS)” to evaluate the deleteriousness of *MLH1* missense VUS. The method extracts protein structural information by using the Ramachandran plot-molecular dynamics simulation (RP-MDS) method, then combines the variation data with an unsupervised learning model composed of auto-encoder and neural network classifier to identify the variants causing significant change in protein structure. In this report, we applied the method to classify 447 *MLH1* missense VUS. We predicted 126/447 (28.2%) *MLH1* missense VUS were deleterious. Our study demonstrates that DL-RP-MDS is able to classify the missense VUS based solely on their impact on protein structure.

## 1. Introduction

MutL protein homolog 1 (MLH1) plays a crucial role in repairing the mismatched DNA due to DNA replication errors [1,2]. Deficiency of MLH1 functionality caused by pathogenic variation can lead to microsatellite instability and a high risk of cancer development as represented by the Lynch syndrome (LS), a hereditary colorectal cancer [3,4], that *MLH1* is mutated in approximately half of the LS cases [3,4]. Identification of *MLH1* pathogenic variant carriers is critical for the prevention and early diagnosis of cancer. Efforts in the past decades have identified many germline variants in human *MLH1.*

Among different types of genetic variations in *MLH1* is the missense variant, which causes a single codon change. The functional significance of many missense variants remains not clear but classified as the variants of uncertain significance (VUS). For example, 1880 (34.4%) of the 5468 *MLH1* variants in the ClinVar database were classified as VUS (https://www.ncbi.nlm.nih.gov/clinvar/, accessed on 2 January 2024). The carriers of *MLH1* VUS variants are uncertain about their cancer risk, therefore, are unable to receive appropriate surveillance and treatment actions. However, further classification of VUS variants into either pathogenic or benign variants remains challenging by current arts in functional classification of genetic variants [5], as the abundance of genetic variants accumulated is far beyond the capacity of the experimentally-based functional testing system. The limitation has promoted the use of computation-based in silico methods as a solution to address the problem. Many in silico methods based on different principles have been developed, such as evolution conservation, population statistics, computation, experiments, and familial segregation [6,7,8,9,10,11,12]. Recently, the Machine Learning (ML) and Deep Learning (DL) have also been integrated into many in silico methods [13,14,15]. However, decade’s practice of in silico methods didn’t reach the original expectation as indicated by ACMG/AMG: “while many of the different software programs use different algorithms for their predictions, they have similarities in their underlying basis; therefore, the combination of predictions from different in-silico tools are considered as a single piece of evidence in sequence interpretation as opposed to independent pieces of evidence”, “tend to have low specificity, resulting in overprediction of missense changes as deleterious, and are not as reliable at predicting missense variants with a milder effect” [16]. In particular, balancing sensitivity and specificity remained a serious issue for most in silico methods [17,18]. More efforts based on new principles need to make to explore the power of in silico approaches to interpret functional significance of genetic variants.

Protein structure can largely determine gene function [19,20,21]. Therefore, protein structural instability affected by genetic variation can be a valuable reference to determine the pathogenicity of genetic variation. Based on this concept, we recently developed the “Ramachandran Plot Molecular Dynamics Simulations (RP-MDS)” method, a dynamical protein structural-based method to measure the deleteriousness of missense variants [22,23]. In the method, the Ramachandran plot (RP) maps the influence of the unclassified missense variant on protein structure stability, and the molecular dynamics (MD) simulations simulate the dynamic trajectories of protein atoms under the influence of changed residue by the missense variant [24]. MD simulations and RP were commonly used in many biomedical studies, such as molecular docking for drug docking, and protein-protein interactions [25,26,27]. We used the combination of known benign and pathogenic variants to determine the thresholds for the benign and pathogenic variants. We then used the thresholds to classify the VUS missense variants into deleterious or non-deleterious variants. We further integrated DL with the RP-MDS method to form DL-RP-MDS, in which an unsupervised learning model (auto-encoder) and a multi-layer neural network classifier were used to compress the information-dense RP into low-dimensional latent space for variant classification [28,29]. Here, we used DL to improve the resolution of the non-linear protein structural changes masked within the RP-MDS data. Furthermore, DL-RP-MDS used the Synthetic Minority Oversampling Technique (SMOTE) to heighten the recognition of the minority dataset in the imbalanced benign and pathogenic data [30]. DL-RP-MDS substantially enhances the accuracy and efficiency of RP-MDS for missense variant classification [31].

In the current study, we used an additional 1μs wildtype allele simulation to balance the limited number of benign variants in the DL-RP-MDS model and applied the new model to classify 447 *MLH1* missense VUS. We were able to identify 126 (28.2%) of the 447 missense VUS as deleterious variants.

## 2. Results

### 2.1. Construct Mutant MLH1 Structures

The following *MLH1* variants from the ClinVar database were used in current study: (1) A total of 44 pathogenic variants (42 variants had three stars by expert reviewer panels, 2 variants had two stars with criteria provided by multiple submitters without conflicts); (2) A total of 8 benign with three stars; and (3) A total of 447 VUS missense variants (3 had three stars, 211 had two stars, and 233 had one star with criteria provided by single submitters without conflicts) (Appendix A).

We first used the benign and pathogenic variants to train the DL-RP-MDS in order to determine the thresholds for variant classification. Of the 52 MLH1 missense variants in ClinVar, 2 benign and 29 pathogenic variants were in the ATPase (ATP) domain, 6 benign and 15 pathogenic variants were in the MutS homologs interaction (MutS-HI) domain (Figure 1). Because of the limited number of MLH1 benign variants, we simulated the wildtype MLH1 for 1 μs to compensate for the benign configurations.

MLH1 crystal structure [PBD ID: 4P7A, resolution 2.30 Å, ATP domain (1–207) and MutS-HI domain (208–346)] [32] were used as the templates to build the mutant structures for each missense variant by following the procedures [23]. Briefly, MODELLER in the Chimera package was used to construct the missed atoms and to select the mutant structure model with the lowest zDOPE score [33]. The template residue was replaced with the altered amino acid residue by a given missense variant from the Dunbrack rotamer library [34,35]. Even with the lowest zDOPE score, some residues may have an unfavorable position (close to positive zDOPE). The residues were relaxed to a more favorable position during the MD simulations. The transformed protein was used as the initial structure for MD simulations.

### 2.2. Use Wildtype Structure to Compensate for Limited Benign Structures

The imbalance of benign and pathogenic variants could overfit towards the over-selection of deleterious variants, due to the lack of structural information on benign structures for DL-RP-MDS. Although the Synthetic Minority Oversampling Technique (SMOTE) was used to balance the data, the technique only synthesized new information within the existing sample. Thus, we used the wildtype structure with 1 μs simulation to improve the disproportional structures between benign and pathogenic variants. The 1 μs simulation gave an additional 29,726 structural frames for the DL-RP-MDS training set and increased the possible structural configurations that were unlikely deleterious. The wildtype MD simulation showed that the ATP domain remained stable and had minimal structure fluctuation, although the MutS-HI domain showed certain fluctuation between the α-helix I (residues 269–282) and the α-helix H (residues 321–336) (Figure 2). The violin plot showed the statistical distribution of the wildtype structure root-mean-square-deviation (RMSD). At the RMSD ≈ 0.75 nm, the α-helix I and H compact (closed) position was the preferential state of the domain, whereas other RMSD distances showed weak interactions between α-helix I and α-helix H (open).

### 2.3. Tuning Parameter of DL-RP-MDS

We set up the model’s parameters for DL-RP-MDS to test the wildtype allele, benign and pathogenic variants. The MD simulations generated the Ramachandran scatters plot (RSP) for each wildtype allele, benign, and pathogenic variants and submitted the RSP to the autoencoder (AE). The AE treated wildtype and benign variant RSP as “benign”. Using the procedures in the ref [31], we determined that the optimal hyperparameter configuration for the classifier was a multi-layer neural network with three hidden layers M″=3, each with 512 nodes and no dropout, together with a latent representation dimension of *q* = 6 and pseudo seed = 0 (Figure 3). Thus, we employed six different dimensions to analyze the relationship between benign and pathogenic variants. We further validated the model by repeating five times of the four-fold stratified cross-validation and achieved 1.00 and 1.00 for the testing and training data for MLH1 (Appendix A). Therefore, the results demonstrated that the DL-RP-MDS model was not overfitted.

The partially overlapped regions between benign variants and wildtype alleles confirmed the similarity between the benign and wildtype structure configurations (Figure 4). Conversely, the distinct region where the wildtype alleles did not overlap with the benign variants was assumed to be the region of benign structural features. The partially overlapped region between the benign/wildtype and pathogenic variants reflected the common features of the mutant structure. Non-overlapped regions represented the distinct protein deformities caused by the pathogenic missense variants. The results of the individual variants were presented as the probability of “deleterious” (P(D)) and “unknown” (P(U)) (Appendix A). The variants with a high probability (>85%) were defined as having high prediction confidence, while the prediction confidence diminished as the variants approached the probability of 50%.

### 2.4. Classification of VUS Variants by DL-RP-MDS

Using the DL-RP-MDS model, we analyzed the 447 *MLH1* missense VUS variants. DL-RP-MDS predicted 126 variants (28.2% of the 447 variants) as deleterious, of which 81 were located in the ATP and 45 in MutS-HI domain (Appendix A). We used a two-sample *t*-test to compare the distribution of the deleterious VUS (from our prediction) and pathogenic variants (from ClinVar database) in the ATP and MutS-HI domains. We observed no significant differences between the deleterious VUS and pathogenic variants (*p*-value = 0.12 and 0.33 for ATP and MutS-HI domains, respectively), which signified that DL-RP-MDS was able to determine deleterious VUS within the distribution of pathogenic variants (Appendix A). DL-RP-MDS identified 290 VUS as deleterious with the training set based on 8 benign and 45 pathogenic variants, whereas the additional wildtype structure reduced the number of deleterious variants to 126. The wildtype expanded the area common structure in the latent dimension (Figure 4).

Table 1 showed the variants that were determined to have deleterious probability >0.85 (Figure 5, Appendix A), and these variants had caused a structural change, therefore, likely to cause a functional change in the protein.

We measured the root mean square deviation (RMSD), root mean square fluctuation (RMSF), solvent accessible surface area (SASA), and radius of gyration of the deleterious VUS (Appendix A). RMSD showed large fluctuation between different variants within the 40 ns time frame, implying the less stable structure; RMSF showed large fluctuation between 300–347 residues; SASA and radius of gyrate showed relatively stable structure during simulation process. The combination of the results highlighted that most variants affected local but not global structures. For example, G181D caused significant structural change [p(D) > 0.95 confidence]: The wildtype G181 residue interacted with 4 residues (L177, S184, V185, and I219) that L177, S184, and V185 residues were in the αF helix, I219 was in the αG helix, and all had limited interactions with the neighbor residue branches (Figure 6a), whereas the altered residue D181 interacted with 5 residues (L177, E178, S184, V185, and I219). The aspartic acid residue branch had multiple interactions with the I219 residue branch, causing structural instability of αG (Figure 6b). Another example is V326M [p(D) > 0.98 confidence]: the wildtype V326 residue interacted with 9 residues (F240, M242, L272, I322, N329, I330, S340, R341, and M342) (Figure 6c), and the wildtype V326 residue branch interacted with F240 and M242 in the β10 strand, which stabilized the αH on the β sheet. The variant residue M326 also interacted with 9 residues (L272, I322, L323, R325, Q328, I330, S340, M342, and Y343), of which 4 were different from 9 residues the wildtype V326 interacted with, and L272, I322, and I330 residue branches, which destabilized the spatial position of αI and αH. Furthermore, the wildtype backbone interacted with the neighboring backbone to maintain the αl helix structure. The variants residue lost the interaction with the β sheet but created new interactions with L323, R325, Q328 within the αI helix, consequently causing the αH to curve and destabilizing the αI helix structure (Figure 6d).

We also applied the 22 in silico methods to predict the same 447 VUS (Table 2 and Appendix A). Except for the Primate_AI’s prediction with 12% (54/447) as deleterious, other 21 methods predicted >60% of the VUS as deleterious, of which 12 methods predicted >80% of VUS as deleterious, and 7 methods (MutationTaster, FATHMM, M_CAP, BayesDel_addAF, BayesDel_noAF, LIST_S2, and fathmm_MKL_coding) predicted >90% of VUS as deleterious. In comparison, DL-RP-MDS models predicted 28.2% (126/447) as deleterious, the lowest among all the in silico methods except Primate AI.

## 3. Discussion

Using our established DL-RP-MDS method, we performed a comprehensive characterization of the impact of 447 *MLH1* missense VUS variants on MLH1 stability. Data from our study showed that 126 missense variants (28.2%) were deleterious.

The functional impact for the majority of the genetic variants remains unclear (https://brcaexchange.org/factsheet, accessed on 29 October 2022). Hence, the computational-based in silico approach is expected to play a significant role in determining the functionality for the uncertain variants, owing to its high-throughput capacity, digital nature, and being backed by different theoretical backgrounds. Indeed, multiple in silico methods have been developed for genetic variant analysis. However, a common limitation of the existing in silico methods is their low specificity of prediction, in particular, over-prediction of deleteriousness as observed by others and our current study (Figure 4, Table 2) [14,17,36]. This contradicts the data from functional assays, which consistently demonstrated that only a small portion of the VUS variants were deleterious [37,38]. Therefore, ACMG guidelines concluded that the current in silico methods “… tend to have low specificity, resulting in overprediction of missense changes as deleterious, and are not as reliable at predicting missense variants with a milder effect” [16]. The statement is reflected in the results, the wildly used in silico methods classified the majority of MLH1 VUS as pathogenic (Table 2).

The high specificity of DL-RP-MDS is reflected by its low rate of deleterious variant classification (Table 2) and its ROC data (Appendix A). This is attributed to the unique features of the DL-RP-MDS method: (1) Using DL and autoencoder to condense the information-dense RSP and to analyze nonlinear molecular interactions/structural information of each protein variant [39,40,41]; (2) DL-RP-MDS is based on protein structure to classify VUS in cancer predisposition genes. We reasoned that a coding-changing deleterious variant should cause instability of protein structure. Therefore, using protein structure as a reference, we should be able to classify the coding-change missense VUS variant into either deleterious or non- deleterious. This approach largely avoids the uncertainty, complexity, and artefacts in other in silico methods. For example, our study indicates that deleterious variants in human cancer genes did not originate from other species through evolution conservation but arose in recent evolution process of human itself [42,43,44]. This implies that the concept of evolution conservation can’t be used to classify deleterious variants in humans; (3) DL-RP-MDS uses MD simulations to monitor the trajectories of structural change caused by VUS variants. The dynamical change of protein conformations can reflect better the instability caused by the altered residue than the experimentally determined static crystal structure [45,46,47]; (4) Using the information of ϕ and Ψ torsional angle to increase the accuracy of structural changes caused by genetic variants [8,13]; (5) the “gene-specific” feature of DL-RP-MDS method. The protein structure for each gene is specific, and the degree of acceptable deviation is independent of other genes. This feature leads to higher sensitivity and specificity of DL-RP-MDS than other in silico methods, which were often designed as universal tools for different genes. (6) DL-RP-MDS provides a probability of “deleterious” and “unknown”, and the final classification results are based on the highest probability of the prediction, avoiding the use of an arbitrarily defined cut-off for the binary classification. The recent success of the structure prediction algorithms (i.e., AlphaFold 2, I-TASSER, and Meta AI) provides the predicted protein structures for many genes currently without protein structure information [48,49,50,51]. Those algorithms also utilize DL and artificial intelligence language models to search for evolutionary patterns in protein sequences, similar protein structures in convergent evolution genes, and atomistic structure alignment to predict novel protein structures. DL-RP-MDS can use the newly predicted structures to classify the deleterious missense variants in different genes. A large portion of missense variants in *MLH1* remains functionally unknown. The data generated by DL-RP-MDS provided MLH1 structure-based evidence to further classify the unknown *MLH1* variants into deleterious or non-deleterious groups, and made a solid step towards determining the unknown *MLH1* variants into clinically actionable Pathogenic or Benign variants for clinical diagnosis and treatment.

Limitations exist in our study: (1) Lack of experimental validation for the results predicted by DL-RP-MDS; (2) The optimization of the multi-layer neural network is non-deterministic. The effects of the initialization order and optimization algorithm may introduce uncertainty into the neural network. Therefore, we introduced the fixed seed and the usage of CPU computations to increase the reproductivity of the results. However, we remained cautious that the P(deleterious, D) or P(unknown, U) may change with each DL training; (3) features chosen for DL classification remained elusive due to dimension reduction by the AE persisted as an unknown property. Therefore further exploration of AE and protein structure relation will be needed [31,52]; (4) DL-RP-MDS was not able to predict the structural change caused by protein-protein interaction; (5) MD simulations may not capture all conformational space and entrap into local minima basins; (6) DL-RP-MDS requires substantial computational power for high-throughput classification [31]. (7) The seed model was chosen based on the lower number of deleterious variants and required further fine-tuning for the selection, such as integrating a soft voting classifier in the DL-RP-MDS. The current design of DL-RP-MDS focused on structural change. However, DL-RP-MDS could be significantly enhanced by integrating protein-binding information as a feature. DL-RP-MDS can be further applied to test the missense VUS variants in other MMR genes such as MSH2 and MSH6 and integrated with experimental evidence such as these from MAVEs (Multiplex assays of variant effect) for validation [53,54].

## 4. Materials and Methods

### 4.1. Source of Missense Variants

A total of 500 missense variants, including 45 pathogenic, 8 benign, and 447 missense VUS variants from the ClinVar database were used in the study (https://www.ncbi.nlm.nih.gov/clinvar/, accessed on 22 October 2020).

### 4.2. Molecular Dynamics Simulation

MD simulations for all missense variants were performed by GROMACS v. 2021 as described in the procedures [55]. Briefly, we extracted the MLH1 N-terminus structure from the PDB database (PDB ID: 4P7A, 0–348 residues, 2.30 Å resolution). All solvent and solvate molecules except Zn^2+^ ions were removed from the PDB file, and the protein was situated at the center of the 10 nm cubic simulation box [32]. Afterward, all simulation boxes were saturated with TIP3P water and neutralized with Cl^−^ ions. We used the AMBER14 force field to model the protein complex with the ions (Zn^2+^ and Cl^−^) [56]. Each simulation box had approximately ~99,000 atoms. The steepest descent algorithm was first applied to the system before 1 ns of equilibration run was performed, with the system condition set as 298 K and 1 bar and utilizing the Berendsen temperature and pressure coupling. Subsequently, a 40 ns production run was simulated for all variants, and a 1 μs production run was simulated for the wildtype allele. The production run utilized V-rescale temperature coupling and Parrinello-Rahman pressure coupling to set the system conditions as 298 K and 1 bar [57]. A time step of 2 fs was utilized for integrating the Verlet velocity algorithm. The Particle Mesh Ewald (PME) method treated the long-range electrostatic potentials, and the cut-off distance was set at 1.0 nm. LINear Constraint Solver (LINC) algorithms were used to constrain hydrogen bonds at equilibrium length [58].

### 4.3. Deep Learning Ramachandran Plot Molecular Dynamic Simulation

GROMACS command “gmx rama” was used to extract Ramachandran scatter plots (RSP) from the MD simulation trajectories [23], and DL-RP-MDS was used for the classification procedure [31]. The last 10 ns of the MD simulation production run was used to generate RSPs and passed to the DL-RP-MDS. Besides the known benign and pathogenic training set used in the previous study [31], we increased the number of benign variants by including wildtype allele simulations and extracted RSPs every 10 ns from 30 ns to 1 μs, in order to balance the number of benign and pathogenic RSP samples. DL-RP-MDS used a time-one-lagged autoencoder (AE) to find the simplified dimensional representation of RSP, retaining the torsional configurations of the MLH1 protein structure. DL-RP-MDS used a time-one-lagged autoencoder (AE) to find the simplified dimensional representation of RSP (latent representation), retaining the torsional configurations of the MLH1 protein structure. The latent representation H is obtained through the *M* hidden layer encoder E which is given by
$$H=E\left(X\right)=W_{M+1}\circ \left(\sigma_{M}\circ W_{M}\right)\circ \cdots \circ \left(\sigma_{1}\circ W_{1}\right)\left(X\right)$$

For input X, weight matrix W and the Leaky-Rectified Linear Unit (ReLU) activation function σ, and the original representation was reconstructed through the decoder D
$$X^{\prime }=D\left(H\right)={W^{\prime }}_{M+1}\circ \left(\sigma_{M}\circ {W^{\prime }}_{M}\right)\circ \cdots \circ \left(\sigma_{1}\circ {W^{\prime }}_{1}\right)\left(H\right).$$

The AE was trained to minimize a reconstruction error, and the mean square error was used as the loss function
$$L\left(X,X^{*}\right)=\frac{1}{N}{\left\|X^{*}-D(E\left(X\right))\right\|}_{2\;}^{2}$$
where X is the input (standardized) atomic configuration at a given time point to be encoded, $X^{*}$ is the torsional configuration of the trained data at the next time point, ${∥\cdot ∥}_{2}$ denotes the $L^{2}$ norm and *N* input data points.

We used the Glorot uniform initialization method to randomly assign values to the weights and biases of the model. To optimize the model parameters, we applied Adam’s algorithm with a learning rate of 0.001 [59]. Afterwards, we performed classification of the latent dimension by using the multi-layer neural network model N which can be expressed as
$$N\left(H\right)=\left({\sigma \prime \prime }_{M^{\prime \prime }+1}\circ {W^{\prime \prime }}_{M^{\prime \prime }+1}\right)\circ \left(\sigma_{M^{\prime \prime }}\circ {W^{\prime \prime }}_{M^{\prime \prime }}\right)\circ \cdots \circ \left(\sigma_{1}\circ {W^{\prime \prime }}_{1}\right)\left(H\right)$$
where only the last activation function σ″ is set to the sigmoid activation function.

The AE was trained to minimize a reconstruction error, and the mean square error was used as the loss function. To optimize the model parameters, we applied Adam’s algorithm with a learning rate of 0.001 [59]. Afterwards, we performed classification of the latent dimension by using the multi-layer neural network model. We connected the neural network model with hidden layers as multi-label classification model. The model mapped the inputs into two classification labels of deleterious or unknown, representing the probability confidence level.

We used “Seed” numbers ranging from 0 to 10,000 with a bin size of 100 to determine the optimal pseudo number for the DL-RP-MDS model. The optimal seed number was determined by choosing the lowest number of deleterious VUS to reduce the probability of overpredicting variants as deleterious. Wildtype allele, benign and pathogenic variants in ClinVar were used as the controls to generate the model for missense VUS classification. Classification of the variants was based on the highest probability of the prediction, i.e., P(unknown, U) and P(deleterious, D).

Balanced accuracy (BA) of DL-RP-MDS was calculated by referring to the average sensitivity and specificity of the variants classified by the model:$$balanced\;accuracy=\frac{sensitivity+specificity}{2}$$

A four-fold stratified cross-validation test with five repeats was conducted to assess the sensitivity and specificity. A total of 20 models were created for DL-RP-MDS, and the ROC and AUC were used to reflect the model performance. For each variant, the 334 frames of RSP were treated as an individual sample, and randomized permutation was undertaken before stratified sampling. ROC curve was calculated by using the Python library Scikit-learn.

The main modification in current study from our established model was the use of an additional 1 μs wildtype allele simulation to balance the limited number of benign variants in the DL-RP-MDS model.

## 5. Conclusions

Using the DL-RP-MDS method, we were able to classify 126 MLH1 missense VUS as deleterious variants from 447 *MLH1* missense VUS. Applying the known benign and pathogenic variants and 1 µs wildtype simulation significantly increases the specificity of DL-RP-MDS for identifying the non-deleterious variants, and avoids the inherited problem of overpredicting deleteriousness inherited in current in silico methods. A combination of the structure-delivered information with experimentally derived functional data and clinical information should promote the further determination of deleterious variants into clinically actionable pathogenic variants for clinical applications.

## Figures and Tables

**Figure 1 ijms-25-00850-f001:**
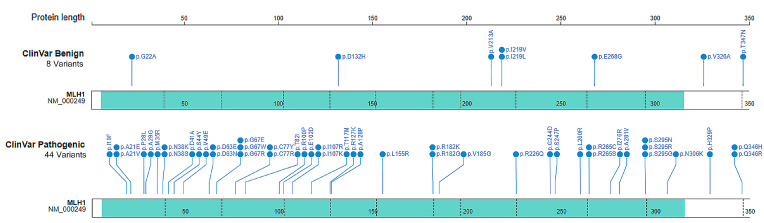
Distribution of benign and pathogenic variants in the MLH1 N-terminus (residues 1–347). The locations of benign and pathogenic variants were marked. The benign variants were enriched in residues 18–141 and 213–347, whereas the pathogenic variants were enriched in residues 19–155 and 226–346. Blue: variant position.

**Figure 2 ijms-25-00850-f002:**
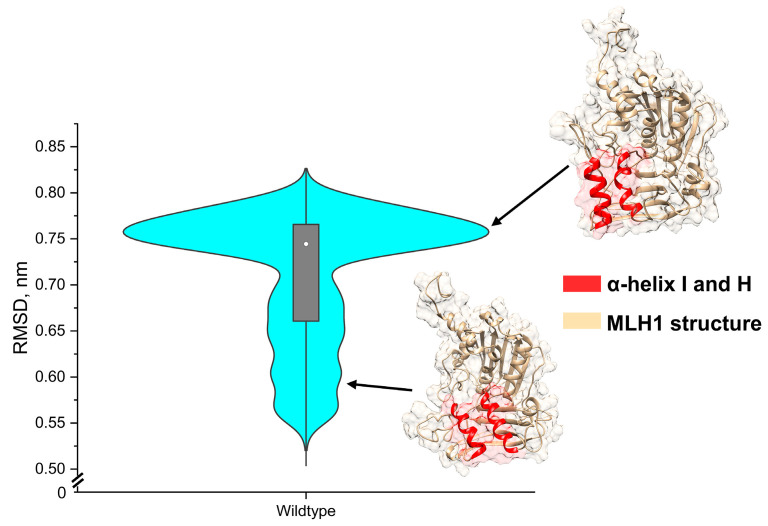
The violin distribution plot of 1μs wildtype RMSD. The MutS-HI domain caused the most significant change in RMSD. At 0.75 nm, the α-helix I and H in the MutS-HI domain showed a more compact (closed) structure. Whereas the other RMSD distance, the α-helix I and H, dissociated with higher structure flexibility. Red: the α-helix I and H; Tan: structure of MLH1; Teal: distributions of RMSD.

**Figure 3 ijms-25-00850-f003:**
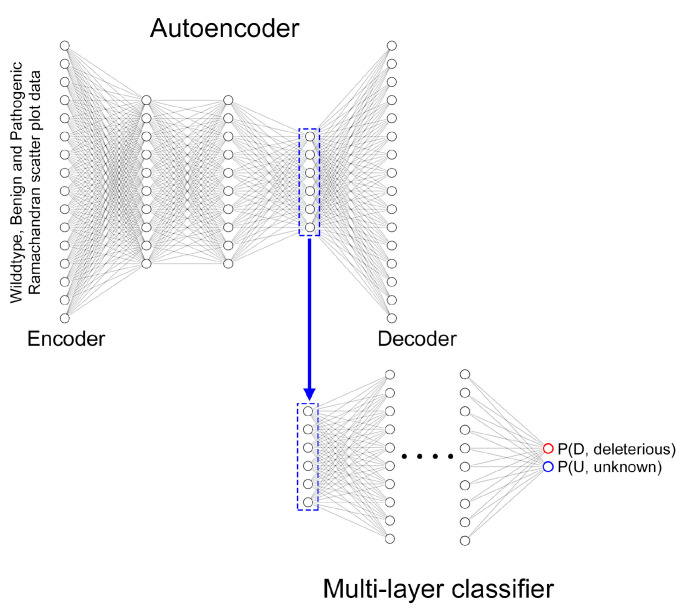
Illustration of an autoencoder and multi-layer classifier. DL-RP-MDS used the wildtype, benign, and pathogenic Ramachandran scatter plot as the input data. The model was optimized with three hidden layers in the autoencoder. The six latent representation dimensions were used in the multi-layer classifier, and the model output the probability of deleterious and unknown for the variants.

**Figure 4 ijms-25-00850-f004:**
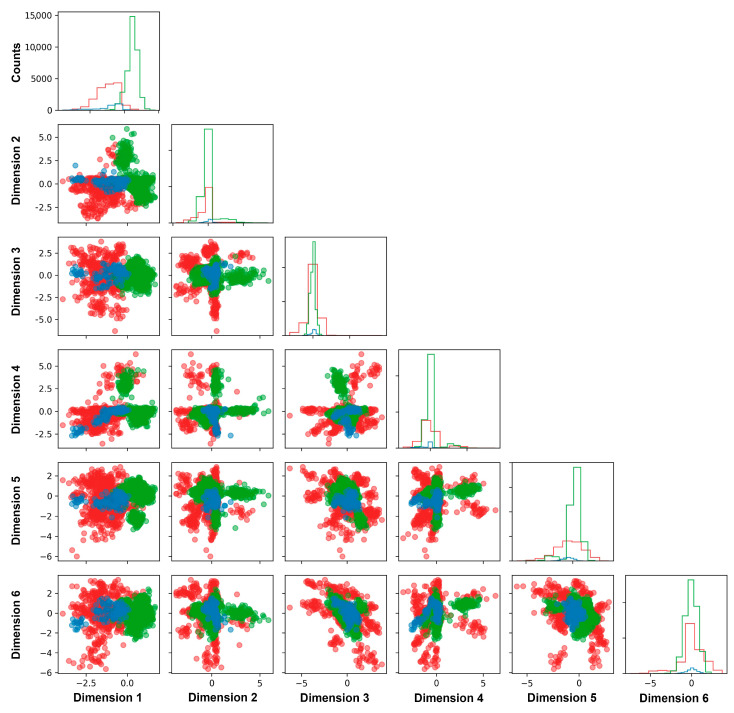
Latent dimensions generated by DL-RP-MDS model. Ramachandran scatter plots were reduced by autoencoder and retained dense-information dimensions. The overlapped regions between benign, WT, and pathogenic variants showed the common structural features of the variants, whereas the non-overlapped regions represented the structural features caused by the variants. The criteria for classification were based on the combination of each latent dimension. Blue: Benign; Red: Pathogenic; Green: wildtype allele.

**Figure 5 ijms-25-00850-f005:**
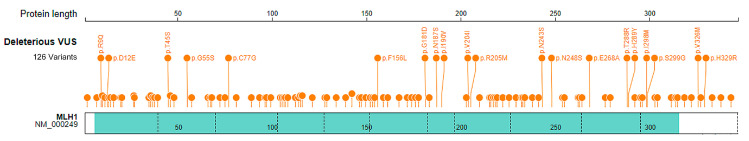
Distribution of deleterious VUS variants by DL-RP-MDS model. The orange lollipop represents the location of predicted deleterious variants in MLH1. Deleterious variants with p(D) > 85% were named in the figure. Orange: deleterious variant position.

**Figure 6 ijms-25-00850-f006:**
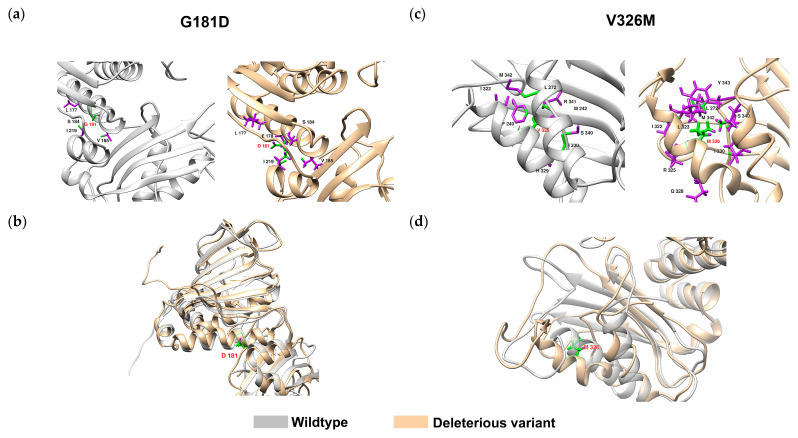
Structural change in MLH1 by G181D and V326M. (**a**) The wildtype G181 interacted with 4 residues (L177, S184, V185, and I219), whereas the variant D181 interacted with 5 residues (L177, E178, S184, V185, and I219). (**b**) The D181 caused instability of αG and further affected the MutS-HI domain. (**c**) The wildtype V326 interacted with 9 residues (F240, M242, L272, I322, N329, I330, S340, R341, and M342), whereas the M326 interacted with altered 9 residues (L272, I322, L323, R325, Q328, I330, S340, M342, and Y343). (**d**) The M326 did not interact with the β sheet and caused the αI helix to detach. Grey: wildtype; peach: variant; green: interacting atoms; purple: non-interacting atoms; red label: wildtype and variants; black label: interacting residues.

**Table 1 ijms-25-00850-t001:** VUS with the highest (>0.85) deleterious probability.

Genome Position (GRCh37)	Change		Location	DL-RP-MDS	
	Nucleotide	Amino Acid		P(U)	P(D)
chr3:37061892	c.976G>A	p.V326M	αI	0.02	0.98
chr3:37050393	c.542G>A	p.G181D	αF	0.05	0.95
chr3:37061902	c.986A>G	p.H329R	αI	0.06	0.94
chr3:37053527	c.614G>T	p.R205M	β8	0.06	0.94
chr3:37050317	c.466T>C	p.F156L		0.07	0.93
chr3:37042467	c.229T>G	p.C77G		0.09	0.91
chr3:37053333	c.568A>G	p.I190V		0.09	0.91
chr3:37059069	c.863C>G	p.T288R		0.10	0.90
chr3:37061811	c.895A>G	p.S299G		0.11	0.89
chr3:37055988	c.743A>G	p.N248S		0.11	0.89
chr3:37038126	c.133A>T	p.T45S		0.12	0.88
chr3:37035074	c.36C>G	p.D12E	αA	0.12	0.88
chr3:37035064	c.26G>A	p.R9Q		0.12	0.88
chr3:37055973	c.728A>G	p.N243S	β10	0.13	0.87
chr3:37061810	c.894C>G	p.I298M	β12	0.14	0.86
chr3:37059071	c.865C>T	p.H289Y		0.15	0.85
chr3:37059009	c.803A>C	p.E268A		0.15	0.85
chr3:37053325	c.560A>G	p.N187S	αF	0.15	0.85
chr3:37038156	c.163G>A	p.G55S		0.15	0.85
chr3:37053523	c.610G>A	p.V204I	β8	0.15	0.85

**Table 2 ijms-25-00850-t002:** VUS classification by different in silico methods.

Methods	Deleterious Variants *	Total Deleterious (%)	Tolerate Variants **	Total Tolerate (%)
MutationTaster	440	98.4	7	1.6
fathmm_MKL_coding	436	97.5	11	2.5
BayesDel_addAF	431	96.4	16	3.6
M_CAP	430	96.2	17	3.8
FATHMM	426	95.3	21	4.7
BayesDel_noAF	418	93.5	29	6.5
LIST_S2	414	92.6	33	7.4
ClinPred	393	87.9	54	12.1
DEOGEN2	384	85.9	63	14.1
LRT	381	85.2	66	14.8
REVEL	380	85.0	67	15.0
fathmm_XF_coding	358	80.1	89	19.9
MetaRNN	356	79.6	91	20.4
SIFT	316	70.7	131	29.3
MetaLR	301	67.3	146	32.7
PROVEAN	290	64.9	157	35.1
MutationAssessor	285	63.8	162	36.2
SIFT4G	282	63.1	165	36.9
MetaSVM	281	62.9	166	37.1
Polyphen2_HDIV	247	55.3	200	44.7
Polyphen2_HVAR	213	47.7	234	52.3
DL-RP-MDS	126	28.2	321	71.8
PrimateAI	54	12.1	393	87.9

* Deleterious in all methods except Polyphen2_HDIV/Polyphen2_HVAR: Probably damaging and Possible damaging, MutationTaster: Disease causing, ** Tolerate in all methods except Polyphen2_HDIV/Polyphen2_HVAR: Benign; LRT/MutationAssessor/PROVEAN/fathmm-MKL_coding: Neutral; MutationTaster: Polymorphism.

## Data Availability

The data presented in this study are openly available in https://doi.org/10.5281/zenodo.7435215, accessed on 28 June 2023 for DL-RP-MDS software and https://doi.org/10.6084/m9.figshare.22785491.v1, accessed on 11 May 2023 for all MLH1 dataset.

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
