# Peer review of "Classification of MLH1 Missense VUS Using Protein Structure-Based Deep Learning-Ramachandran Plot-Molecular Dynamics Simulations Method"

_ijms, 2024, doi:10.3390/ijms25020850_

Round 1

Reviewer 1 Report

Comments and Suggestions for Authors

This manuscript presents a numerical study on the development of models for the Classification of MLH1 missense VUS using protein structure data. This work is rather interesting and can be considered for publication after some modifications are carried out in the manuscript.

There is a lack of sufficient literature review on the specific subject. The authors should add at least two paragraphs, in order to present the most important works on this subject, including recent ones in order to prove their novelty.

All abbreviations should explained in the manuscript.

The sentence "The weight and biases We used the Glorot uniform initialization method to randomly assign values to the weights and biases of the model." should be appropriately corrected.

More details about the deep learning model, including a relevant schematic, are necessary.

The various "in silico" methods presented in Table 1 should be briefly explained in the manuscript in order to better clarify their characteristics in comparison to the proposed model. How does the DL-RP-MDS method differ from existing in silico methods for genetic variant analysis, and what are the advantages of this approach?

What are the potential implications of the study's findings for the diagnosis and treatment of Lynch syndrome and other hereditary cancers, and how might the DL-RP-MDS method contribute to personalized medicine?

Author Response

Question

There is a lack of sufficient literature review on the specific subject. The authors should add at least two paragraphs, in order to present the most important works on this subject, including recent ones in order to prove their novelty.

Answer

We thank the reviewer for the suggestion. We have additional paragraphs in the introduction to explain the need for further in silico methods and expand the challenge of classifying missense variants:

… Among different types of genetic variations in MLH1 is the missense variant, which causes a single codon change. The functional significance of many missense variants remains not clear but classified as the variants of uncertain significance (VUS). For example, 1,880 (34.4%) of the 5,468 MLH1 variants in the ClinVar database were classified as VUS (https://www.ncbi.nlm.nih.gov/clinvar/, accessed January 2, 2024). The carriers of MLH1 VUS variants are uncertain about their cancer risk, therefore, are unable to receive appropriate surveillance and treatment actions. However, further classification of VUS variants into either pathogenic or benign variants remains challenging by current arts in functional classification of genetic variants [6], as the abundance of genetic variants accumulated is far beyond the capacity of the experimentally-based functional testing system. The limitation has promoted the use of computation-based in silico methods as a solution to address the problem. Many in silico methods based on different principles have been developed, such as evolution conservation, population statistics, computation, experiments, and familial segregation [7-13]. Recently, the Machine Learning (ML) and Deep Learning (DL) have also been integrated into many in silico methods [14-17]. However, decade’s practice of in silico methods didn’t reach the original expectation as indicated by ACMG/AMG: “while many of the different software programs use different algorithms for their predictions, they have similarities in their underlying basis; therefore, the combination of predictions from different in-silico tools are considered as a single piece of evidence in sequence interpretation as opposed to independent pieces of evidence”, “tend to have low specificity, resulting in overprediction of missense changes as deleterious, and are not as reliable at predicting missense variants with a milder effect". In particular,  balancing sensitivity and specificity remained an serious issue for most in silico methods [18, 19]. More efforts based on new principles need to make to explore the power of in silico approaches to interprate functional significance of genetic variants.

Question

All abbreviations should explained in the manuscript.

Answer

In the revision, we have included full names for all abbreviations.

Question

The sentence "The weight and biases We used the Glorot uniform initialization method to randomly assign values to the weights and biases of the model." should be appropriately corrected.

Answer

We have revised the sentence in the revision:

We used the Glorot uniform initialization method to randomly assign values to the weights and biases of the model.

Question

More details about the deep learning model, including a relevant schematic, are necessary.

Answer

We thank the reviewer for this comment. We have expanded the methods and added a schematic for the autoencoder and multi-layer classifier, as figure 3.  We have also added the following sentences in Methods in the revision:

DL-RP-MDS used a time-one-lagged autoencoder (AE) to find the simplified dimensional representation of RSP (latent representation), retaining the torsional configurations of the MLH1 protein structure. The latent representation  is obtained through the M hidden layer encoder  which is given by

for input , weight matrix  and the Leaky-Rectified Linear Unit (ReLU) activation function , and the original representation was reconstructed through the decoder

The AE was trained to minimize a reconstruction error, and the mean square error was used as the loss function

where  is the input (standardized) atomic configuration at a given time point to be encoded,  is the torsional configuration of the trained data at the next time point,  denotes the  norm and N input data points.

We used the Glorot uniform initialization method to randomly assign values to the weights and biases of the model. To optimize the model parameters, we applied Adam's algorithm with a learning rate of 0.001 [27]. Afterwards, we performed classification of the latent dimension by using the multi-layer neural network model  which can be expressed as

where only the last activation function  is set to the sigmoid activation function.

Question

The various "in silico" methods presented in Table 1 should be briefly explained in the manuscript in order to better clarify their characteristics in comparison to the proposed model. How does the DL-RP-MDS method differ from existing in silico methods for genetic variant analysis, and what are the advantages of this approach?

Answer

We thank the reviewer for this comment. As mentioned above, we further described the features of other in silico methods and DL-RP-MDS in the Discussion to distinguish the difference between other in silico methods and DL-RP-MDS:

…2) DL-RP-MDS is based on protein structure to classify VUS in cancer predisposition genes. We reasoned that a coding-changing deleterious variant should cause instability of protein structure. Therefore, using protein structure as a reference, we should be able to classify the coding-change missense VUS variant into either deleterious or non- deleterious. This approach largely avoids the uncertainty, complexity, and artefacts in other in silico mathods. For example, our study indicates that deleterious variants in human cancer genes did not originate from other species through evolution conservation but arose in recent evolution process of human itself [47-49]. This implies that the concept of evolution conservation can’t be used to classify deleterious variants in humans; 3) DL-RP-MDS uses MD simulations to monitor the trajectories of structural change caused by VUS variants. The dynamical change of protein conformations can reflect better the instability caused by the altered residue than the experimentally determined static crystal structure [50-52]; 4) Using the information of ϕ and Ψ torsional angle to increase the accuracy of structural changes caused by genetic variants [9, 14]; 5) the “gene-specific” feature of DL-RP-MDS method. The protein structure for each gene is specific, and the degree of acceptable deviation is independent of other genes. This feature leads to higher sensitivity and specificity of DL-RP-MDS than other in silico methods, which were often designed as universal tools for different genes.

Question

What are the potential implications of the study's findings for the diagnosis and treatment of Lynch syndrome and other hereditary cancers, and how might the DL-RP-MDS method contribute to personalized medicine?

Answer

We thank the reviewer for the additional suggestions. In the revision, we have included the following sentences to address the issue:

A large portion of missense variants in MLH1 remains functionally unknown. The data generated by DL-RP-MDS provided MLH1 structure-based evidence to further classify the unknown MLH1 variants into deleterious or non-deleterious groups, and made a solid step towards determining the unknown MLH1 variants into clinically actionable Pathogenic or Benign variants for clinical diagnosis and treatment.

Reviewer 2 Report

Comments and Suggestions for Authors

The authors of the manuscript titled “Classification of MLH1 Missense VUS Using Protein Structure Based Deep Learning-Ramachandran Plot-Molecular Dynamics Simulations Method” developed a protein structure-based method named “Deep Learning-Ramachandran Plot-Molecular Dynamics Simulation (DL-RP-MDS)” to evaluate the deleteriousness of MLH1 missense VUS.

The topic is interested and overall the manuscript is written well.

I have only few comments

1-     The figures need to be improved, all the figures are bad resolution and unclear.

2-     The authors need to represent some of molecular dynamic results, for example RMSD, RMSF, radius of gyration and SASA and represent these in figures.

3-     The conclusion need to be improved and included more data, it is very short.

4-     The English is okay, but needs proof editing for example line 52 “such as drug docking, protein-protein interactions [12-14].” The author should add and

5-     Also this sentence is unclear “We used the combination to train with known benign and known pathogenic variants and determine the thresholds for the benign and pathogenic variants and then is used to classify the VUS missense variants into deleterious and non-deleterious variants”

Comments on the Quality of English Language

The manuscript needs proof editing, authors need to add "and", "the" 

Author Response

Question

The figures need to be improved, all the figures are bad resolution and unclear.

Answer

We thank the reviewer for this comment. We have increased all images to 600 dpi to increase the resolution.

Question

The authors need to represent some of molecular dynamic results, for example RMSD, RMSF, radius of gyration and SASA and represent these in figures.

Answer

We thank the reviewer for this comment. In the revision, we have added RMSD, RMSF, radius of gyration and SASA for the highest (>0.85) variants as supplementary figure 2-5. In addition, we have added the following sentences in Results to descript RMSD, RMSF, radius of gyration and SASA results:

We measured the root mean square deviation (RMSD), root mean square fluctuation (RMSF), solvent accessible surface area (SASA), and radius of gyration of the deleterious VUS (Figure S2-5). RMSD showed large fluctuation between different variants within the 40 ns time frame, implying the less stable structure; RMSF showed large fluctuation between 300 – 347 residues; SASA and radius of gyrate showed relatively stable structure during simulation process. The combination of the results highlighted that most variants affected local but not global structures.

Question

The conclusion need to be improved and included more data, it is very short.

Answer

We thank the reviewer for this comment. We have expand the conclusion to the following:

Using the DL-RP-MDS method, we were able to classify 126 MLH1 missense VUS as deleterious variants from 447 MLH1 missense VUS. Applying the known benign and pathogenic variants and 1µs wildtype simulation significantly increases the specificity of DL-RP-MDS for identifying the non-deleterious variants, and avoids the inherited problem of overpredicting deleteriousness inherited in current in silico methods. A combination of the structure-delivered information with experimentally derived functional data and clinical information should promote the further determination of deleterious variants into clinically actionable pathogenic variants for clinical applications.

Question

The English is okay, but needs proof editing for example line 52 "such as drug docking, protein-protein interactions [12-141 The author should add and

Answer

The sentences have been revised as:

MD simulations and RP were commonly used in many biomedical studies, such as molecular docking for drug docking, and protein-protein interactions.

Question

Also this sentence is unclear “We used the combination to train with known benign and known pathogenic variants and determine the thresholds for the benign and pathogenic variants and then is used to classify the VUS missense variants into deleterious and non-deleterious variants”

Answer

We have revised the sentence as:

We used the combination of known benign and pathogenic variants to determine the thresholds for the benign and pathogenic variants. We then used the thresholds to classify the VUS missense variants into deleterious or non-deleterious variants.

Question

Comments on the Quality of English Language: The manuscript needs proof editing, authors need to add "and", "the"

Answer

For the final version of the revision, we had a native English-speaking colleague to provide proof-reading to improve the quality of English writing of the manuscript.

Round 2

Reviewer 1 Report

Comments and Suggestions for Authors

The authors conducted most of the necessary modifications to their manuscript. Thus, it can now be considered for publication.